# Clinical Characterization and Predictive Factors for Progression in a Cohort of Patients with Interstitial Lung Disease and Features of Autoimmunity: The Need for a Revision of IPAF Classification Criteria

**DOI:** 10.3390/medicina59040794

**Published:** 2023-04-19

**Authors:** Francesco Bozzao, Paola Tomietto, Elisa Baratella, Metka Kodric, Rossella Cifaldi, Rossana Della Porta, Ilaria Prearo, Silvia Maria Grazia Pirronello, Paola Confalonieri, Barbara Ruaro, Fabio Fischetti, Bruno Fabris

**Affiliations:** 1Internal Medicine Department, Azienda ULSS 2 “Marca Trevigiana”, 31100 Treviso, Italy; 2Internal Medicine Department, Rheumatology Unit, Azienda Sanitaria Universitaria Giuliano Isontina (ASUGI), 34128 Trieste, Italy; 3Institute of Radiology, Department of Medical Surgical and Health Sciences, Cattinara Hospital, University of Trieste, 34128 Trieste, Italy; 4Department of Medicine, Surgery and Health Sciences, University of Trieste, 34128 Trieste, Italy; 5Pneumology Unit, Azienda Sanitaria Universitaria Giuliano Isontina (ASUGI), 34128 Trieste, Italy; 6Vascular Medicine Unit, University Hospital LMU Munich, 81377 Munich, Germany; 7Emergency Medicine Unit, Azienda Ospedaliera Cannizzaro, 95126 Catania, Italy; 8Internal Medicine Department, Azienda Sanitaria Universitaria Giuliano Isontina (ASUGI), 34128 Trieste, Italy

**Keywords:** interstitial lung disease (ILD), connective tissue diseases (CTD), interstitial pneumonia with autoimmune features (IPAF), “usual” interstitial pneumonia with autoimmune features (UIPAF), progressive ILD

## Abstract

*Background and Objectives*: The “interstitial pneumonia with autoimmune features” (IPAF) criteria have been criticized because of the exclusion of usual interstitial pneumonia (UIP) patients with a single clinical or serological feature. To classify these patients, the term UIPAF was proposed. This study aims to describe clinical characteristics and predictive factors for progression of a cohort of interstitial lung disease (ILD) patients with at least one feature of autoimmunity, applying criteria for IPAF, specific connective tissue diseases (CTD), and a definition of UIPAF when possible. *Methods:* We retrospectively evaluated data on 133 consecutive patients with ILD at onset associated with at least one feature of autoimmunity, referred by pulmonologists to rheumatologists from March 2009 to March 2020. Patients received 33 (16.5–69.5) months of follow-up. *Results:* Among the 101 ILD patients included, 37 were diagnosed with IPAF, 53 with ILD-onset CTD, and 11 with UIPAF. IPAF patients had a lower prevalence of UIP pattern compared to CTD-ILD and UIPAF patients (10.8% vs. 32.1% vs. 100%, *p* < 0.01). During the follow-up, 4 IPAF (10.8%) and 2 UIPAF (18.2%) patients evolved into CTD-ILD. IPAF patients presented features not included in IPAF criteria, such as sicca syndrome (8.1%), and were more frequently affected by systemic hypertension (*p* < 0.01). Over one year, ILD progression (greater extent of fibrosis on HRCT and/or decline in PFTs) was less frequent in the IPAF group compared to CTD-ILD and UIPAF (32.3% vs. 58.8% vs. 72.7, *p* = 0.02). A UIP pattern and an IPAF predicted a faster (OR: 3.80, *p =* 0.01) and a slower (OR: 0.28, *p* = 0.02) ILD progression, respectively. *Conclusions*: IPAF criteria help identify patients who might develop a CTD-ILD, even though a single clinical or serological feature is respected. Future revisions of IPAF criteria should include sicca syndrome and separate UIP-pattern into a different definition (UIPAF), given its association with a different prognosis, independently from ILD classification.

## 1. Introduction

Connective Tissue Diseases (CTDs) are associated with a range of pulmonary manifestations, of which Interstitial Lung Disease (ILD) is the most prevalent [1,2,3]. ILD often appears early in the course of CTD, and may even be the first manifestation of the disorder in approximately 15% of cases [4,5]. Furthermore, some ILD patients show clinical or serological features of autoimmunity but do not meet diagnostic criteria for any CTD [5,6]. To classify these patients, a task force of the European Respiratory Society (ERS) and the American Thoracic Society (ATS) proposed the term “Interstitial Pneumonia with Autoimmune Features” (IPAF). These criteria are organized around three domains: a clinical domain consisting of symptoms and signs specific for a CTD; a serologic domain consisting of specific autoantibodies; and a morphologic domain consisting of specific chest imaging, histopathologic, or pulmonary pathologic features. To classify a patient as IPAF, known causes for ILD must be excluded and at least one feature from at least two of the domains must be present [6].

So far, there is no consensus about the utility of IPAF criteria in characterizing ILD patients with autoimmune features. In fact, since its publication in 2015, IPAF criteria have been criticized for several aspects, and revisions have been proposed by experts in this field [7,8].

The clinical domain does not include some important, even though nonspecific, features such as sicca syndrome, muscle weakness, and myalgia, often associated with CTD-ILD. It is well-recognized that ILD could precede myositis or sicca syndrome and that ILD patients with myalgia had a high proportion of positivity for Myositis Specific and Myositis Associated Antibodies (MSA/MAA) [9,10,11].

The morphological domain does not include the Usual Interstitial Pneumonia (UIP) pattern, although it is commonly observed in various CTD-ILD and is the most frequent pattern in ILD associated to Rheumatoid Arthritis (RA-ILD) [1]. Furthermore, UIP patients with only one domain (clinical and serological) of IPAF criteria evolved into a Systemic Autoimmune Disease (SAD) significantly more often than Idiopathic Pulmonary Fibrosis (IPF) [12,13]. To classify these patients, some authors proposed the terms “Usual” interstitial pneumonia with autoimmune features (UIPAF) and Autoimmune-ILD (AI-ILD) to classify these patients [12,13].

On the other hand, according to the present IPAF criteria, UIP patients might be designated as IPAF in the same way as patients with other patterns (e.g., NSIP), and this might potentially cause bias in the comprehension of the natural history of such cohorts.

The first aim of this study was to compare the clinical characteristics, evolution and prognosis of IPAF patients to those of CTD-ILD and UIPAF patients in our monocentric cohort. Secondly, we evaluated how IPAF criteria might be improved to recruit all ILD patients with autoimmune features and might be modified to acquire a potential prognostic value.

## 2. Materials and Methods

### 2.1. Patients

In this study, we retrospectively evaluated the clinical data of 133 consecutive ILD patients followed by our tertiary center between March 2009 and March 2020. All the patients had ILD as first manifestation and were all referred to rheumatological consultation by pulmonologists trained to identify clinical or laboratory features suggestive of an underlying SAD.

The exclusion criteria for the study were incomplete clinical data and lack of informed consent. We also excluded patients affected by conditions different to CTD-ILD, IPAF, and UIPAF, e.g., sarcoidosis, vasculitides, and IPF with clinical feature due to a different condition (e.g., osteoarthritis) after rheumatological assessment (the so-called “lone” IPF) [14]. Patients with RA-ILD were included in the CTD-ILD cohort.

Both final diagnoses and therapeutic choices were reached through a multidisciplinary team (MDT) discussion as part of a well-established diagnostic workup adopted in our center. The MDT was composed of rheumatologists, pulmonologists, and radiologists specialized in thoracic imaging, and, when necessary, pathologists. 

For all patients, we collected data regarding demographics, comorbidities, smoking habits, clinical and serological features, ILD pattern, and pulmonary function tests (PFTs) at the time of diagnosis and during the follow-up. All data were collected in a database with less than 1% missing data. The study was conducted according to the Declaration of Helsinki.

### 2.2. Diagnostic Process

All patients underwent the same clinical and instrumental workup, which has been adopted in our tertiary center for ILD characterization.

#### 2.2.1. Clinical Evaluation

Pulmonologists and then rheumatologists detected signs and symptoms suggestive of an underlying SAD, considering all those included in the clinical domain of IPAF plus sicca syndrome, myalgia, and muscle weakness. When a Raynaud Phenomenon was suspected, a nailfold capillaroscopy to detect giant capillaries and avascular areas was performed [15]. To improve specificity for patients with arthralgia, we considered inflammatory arthritis as a clinical feature for IPAF only if unexplained by other conditions (e.g., CTD, seronegative spondyloarthritis and microcrystalline arthritis). Similarly, polyarticular morning stiffness ≥1 h was included as an IPAF feature only if associated with elevation of Erythro-Sedimentation Rate (ESR) or C Reactive Protein (CRP). Referred sicca syndrome was investigated through dry eye tests and unstimulated salivary flow rate, and defined according to the latest ACR/EULAR classification criteria for Sjogren Syndrome (SS) [16]. Minor salivary gland biopsy was performed in dubious cases (e.g., negative anti-Ro/SSA) or to exclude secondary forms of sicca syndrome, and it was interpreted according to the latest criteria [17]. Muscle weakness and myalgia were investigated through the measurement of creatine kinase (CK) and aldolase in the blood, electromyography, and/or muscle magnetic resonance imaging (MRI). 

#### 2.2.2. Laboratory

General blood tests were performed, including complete blood count, kidney and liver function tests, CK, aldolase, Lactate Dehydrogenase, ESR, and CRP. In addition to all the antibodies in the serological domain of IPAF criteria, we tested the patients for other myositis-specific and associated antibodies (MSA/MAA), included in our Western Blot kit “Euroimmun” (Euroimmun Italia, Diagnostica medica s.r.l.): Anti-Mi2a, anti-Mi2b, anti-TIF1γ, anti-NXP2, anti-SAE1, anti-KU, anti-SRP, and anti-RO52. Patients were also tested for anti-phospholipid antibodies and anti-neutrophil cytoplasmic autoantibodies (ANCA), detected both in indirect immunofluorescence and enzyme immunoassay.

#### 2.2.3. Chest High-Resolution Computed Tomography (HRCT)

All patients underwent an HRCT performed with 256-row multidetector CT system (Brilliance iCT 256, Philips, Best, The Netherlands) and acquired during a single breath hold at full inspiration, with the patient in a supine position. Technical parameters were as follows: rotation time, 270 ms; beam collimation, 128 × 2 × 0.625 mm; normalized pitch, 0.975; *z*-axis coverage, 160 mm; reconstruction interval, 0.3 mm; section reconstruction thickness, 1 mm; tube voltage, 120 kV; tube current (effective mA), 280–400 depending on patient size; and field of view, 40 cm. CT images were analyzed at standard lung window settings (window level of −600 HU and window width of 2000 HU) and mediastinal window settings (window level 400–500 HU and window width 20–40 HU). Images were evaluated by an expert thoracic radiologist of our MDT to recognize and classify the ILD pattern, according to the most recent guidelines [14,18]. Only patients with sufficient criteria for a probable or definite Usual interstitial pneumonia (UIP) were classified as having a UIP pattern. After a MDT discussion, a lung biopsy was performed in selected cases to confirm the radiological pattern and to guide therapeutic choices. 

#### 2.2.4. Pulmonary Function Tests (PFTs)

We performed spirometry, diffusing capacity of the lungs for carbon monoxide (DLCO) test, and Six Minutes Walking Test (6MWT) in all patients, according to specific guidelines [19,20,21]. In some patients, an artery blood gas sampling was collected to determine the need for supplemental oxygen, at least under exercise.

#### 2.2.5. Transthoracic Echocardiography

It was used to screen for Pulmonary Hypertension (PH). In patients with a high probability of PH, a right heart catheterization was undertaken, unless contraindicated or refused by the patient. Current guidelines were used to define PH [22]. Echocardiography was also used in diagnosing heart failure according to specific guidelines [23].

### 2.3. Patient Classification

From the overall cohort of 133 ILD patients, 32 of them were excluded from our retrospective analysis due to the presence of exclusion criteria: 5 because of lack of informed consent or incomplete clinical data; and 27 owing to ILD without any feature of the IPAF criteria and affected by conditions different from CTD-ILD and UIPAF (see Figure 1). 

All the remaining 101 subjects presented at least one feature of the IPAF criteria. Of these, 53 reached a final diagnosis of CTD-ILD, conforming to the latest specific diagnostic or classification criteria [16,24,25,26,27,28]. Since Antisynthetase Syndrome (AS) is not currently classifiable by validated classification criteria, we decided to consider as IPAF those patients with anti-tRNA synthetase antibodies positivity plus a non-UIP pattern. The patients with at least one of the other manifestations of the disease, such as myositis or arthritis, were classified as having a definite diagnosis of AS [29]. Of the remaining patients, 37 were classified as IPAF because of one or more features from at least two of the IPAF domains [6]. Of these, 8 were initially classified as lung-dominant undifferentiated CTD (UCTD) before the publication of IPAF criteria in 2015, and they all presented sufficient criteria to be classified as IPAF [30]. Finally, 11 patients had a probable or definite UIP pattern plus one feature in the serological or clinical domain of the IPAF classification criteria. Since they did not satisfy any criteria for specific CTD and IPAF, we classified them as “usual” interstitial pneumonia with autoimmune features (UIPAF), as in Sambataro et al. [13].

### 2.4. Follow-Up

All patients underwent regular pulmonologist and rheumatologist visits, PFTs at least every 6 months, and HRCT at least yearly. Periodic blood tests were used to screen for adverse effects of the prescribed therapies. The autoimmune panel was repeated in patients with borderline autoantibodies positivity or new rheumatologic manifestations. 

The median follow-up was 33 (16.5–69.5) months from the diagnosis. Data were collected at baseline, at 12 ± 3 months from diagnosis (T1) and at the last follow-up visit.

The following clinical and instrumental parameters defined ILD progression at T1:Deterioration of lung function on PFTs, defined by an absolute decline in forced vital capacity (FVC) of ≥5% predicted or in DLCO of 10% predicted [31].Radiological progression on HRCT: the trend of ILD (stable, deteriorated, or ameliorated) was established by the most expert thoracic radiologist, blinded to the clinical and functional evaluation, through a semi-quantitative analysis. Particularly evaluated were the presence or increase of the extent of traction bronchiectasis/bronchiolectasis, ground glass opacities, reticulation, and honeycombing as well as the increase of the lobar volume loss [31]. Figure 2 reports an example of ILD progression on HRCT.New chronic need for supplemental oxygen.Death due to ILD.Acute exacerbations, determining an acute respiratory insufficiency and/or a hospital admission and/or a broad-spectrum antibiotic therapy.

Finally, to evaluate through multivariate analysis predictive factors for lung deterioration, we determined a composite endpoint of ILD progression at T1, considering as deteriorated those patients with a decline in PFTs (defined as reported above) or with a greater extent of fibrosis on HRCT. 

On the other hand, we considered as ameliorated on PFTs those patients with an increase in FVC of ≥10% predicted and/or in DLCO of ≥15% predicted. Patients who did not meet the improved or worsened group criteria were assigned to the “stable” group. Patients with a FVC < 50% predicted or a DLCO < 35% predicted were affected by severe ILD [32,33,34,35].

### 2.5. Statistical Analysis

The analysis was performed through the open-source software “R” (www.r-project.org (accessed on 12 April 2020)).

Categorical variables were reported as counts and percentages. For quantitative variables, we used median and interquartile range (I-III quartiles), considering the non-normal distribution of data through the Shapiro–Wilk test. Hypothesis tests were performed to compare the outcomes between the groups (IPAF, CTD-ILD, and UIPAF). We used the Kruskal–Wallis test to compare change in medians and the Chi-squared test or Fisher’s exact test to compare proportions. If there was a statistically significant difference in overall group comparison, we performed a post-hoc test with Bonferroni at α_c_ = 0.017 to compare individual groups with each other.

Logistic regression was used to determine predictive factors for the composite endpoint of ILD progression at T1, including only parameters with a *p* < 0.1 on univariate analysis in the multivariate models. Results were reported as Odds Ratio (OR) and 95% Confidence Interval (95% CI). Clinical data of deceased patients during the follow-up were excluded from the statistical analysis of the parameters suggestive of ILD progression. 

Survival was evaluated using the Kaplan–Meier survival curves and log-rank test.

A *p*-value < 0.05 was considered statistically significant.

## 3. Results

### 3.1. Baseline Characteristics

In our cohort of 101 patients, 37 were classified as IPAF, 53 as CTD-ILD, and 11 as UIPAF. 

IPAF patients were predominantly female (54.1%), and the median age was 71 (62.5–74.5) years. The most frequently satisfied item of the IPAF domains was the Nonspecific interstitial pneumonia (NSIP) pattern (56.8%), followed by antinuclear antibody (ANA) positivity (43.2%), anti-Ro/SSA positivity (27%), and inflammatory arthritis (24.3%). To be classified as IPAF, the most common combination was morphological and serological domain (48.6%), followed by the combination of all three domains (27%). Some IPAF patients presented clinical or serological features not included in the IPAF criteria. Of these, three patients (8.1%) had objective evidence of sicca syndrome, not explained by other causes, although they did not reach enough criteria for SS at first evaluation [16]. In six IPAF patients (16.2%), we found a high-titer positivity for one or more anti-phospholipid antibodies, confirmed on two occasions, without any definite anti-phospholipid syndrome. Anti-myeloperoxidase (MPO) antibodies were present in one IPAF patient, without any clinical feature suggestive of ANCA-associated vasculitis. Anti-Ro52 antibodies were found in eight patients (21.6%). Further data about patient characteristics are in the Appendix A.

Among the 11 UIPAF patients, eight (72.7%) satisfied the serological domain of the IPAF criteria, mainly for ANA positivity (54.5%). Of the three patients with a clinical feature of the IPAF criteria, two presented Raynaud phenomenon and palmar teleangiectasia, and one arthritis without sufficient criteria for RA. Furthermore, three UIPAF patients (27.3%) had objective evidence of sicca syndrome.

The cohort of CTD-ILD included: 15 Systemic Sclerosis (SSc), 15 RA, 10 SS, eight Poli-/Dermatomyositis, three Systemic Lupus Erythematosus, and two AS. All the patients presented at least one feature of the IPAF criteria, and 24 (45.3%) the combination of all three domains.

No statistically significant difference was found between IPAF, CTD-ILD, and UIPAF regarding median age, sex distribution, smoking habits, and median time range from symptoms’ onset and final diagnoses (see Table 1). The three groups did not differ in the frequency of some comorbidities, except for systemic hypertension and left-sided heart failure, which were most frequent in the IPAF group (*p* < 0.01, *p* = 0.03 respectively). At onset, no UIPAF patient suffered from a concomitant autoimmune disease, in contrast to IPAF and CTD-ILD patients. Of these, autoimmune thyroiditis was the most common, being present in nearly one in five IPAF and CTD-ILD patients. 

IPAF patients were less likely to have a UIP pattern (four patients, 10.8%) compared to CTD-ILD (17 patients, 32.1%) and, as expected, UIPAF patients (*p* < 0.01). Conversely, the NSIP pattern was the most common in the IPAF and CTD-ILD groups. No statistically significant difference was found between the groups in terms of PFTs at onset (see Table 2). 

The prevalence of autoantibody positivity was not significantly different in the three groups. However, the low rate of UIPAF patients with a serological feature interfered with the statistical analysis (see Appendix A). 

Prednisone ≥7.5 mg or equivalent was administered for almost three months in 87.1% of patients. IPAF and CTD-ILD patients were more likely to receive a concurrent immunosuppressant (cyclophosphamide, mycophenolate mofetil, azathioprine, or rituximab), UIPAF an anti-fibrotic agent (pirfenidone or nintedanib). Nevertheless, all IPAF patients with a UIP pattern were treated with an anti-fibrotic therapy, as far as six patients in the CTD-ILD group (10.8%).

### 3.2. Follow-Up

After a median time of 17 (11.3–28.8) months from the diagnosis, six IPAF patients (16.2%) evolved into a different ILD classification. Among these, two patients (5.4%) developed a UIP pattern and were classified as UIPAF, and four (10.8%) evolved into a CTD-ILD. Interestingly, two of the latter patients manifested a sicca syndrome with focal lymphocytic sialadenitis and a focus score of ≥1 foci/4 mm^2^, and SS was diagnosed [16,17]. The clinical characteristics of these patients are in the Appendix A.

During the median follow-up of 33 (16.5–69.5) months, seven IPAF patients (18.9%) developed a neoplasm, of which three had lung cancer, and 18 (48.6%) experienced almost an acute worsening.

In the UIPAF cohort, two patients (18.2%) developed a SAD during the follow-up. The first patient had a doubled rheumatoid factor (RF) and high-titer anti-phospholipid antibody at diagnosis, and reached enough criteria for a concurrent anti-phospholipid syndrome due to intercurrent pulmonary embolism. The other one had ANA positivity and anti-signal recognition peptide (anti-SRP) antibodies, and developed a necrotizing myositis.

The one-year survival of the whole cohort was 95.8%, similar between IPAF, CTD-ILD and UIPAF (*p* = 0.89) (see Figure 3).

At 12 ± 3 months from the diagnosis (T1), the three groups significantly differed in the frequency of patients with a FVC < 50% predicted or a DLCO < 35% predicted, in the need for supplementary oxygen, and in the probability of reaching the composite endpoint of ILD progression (see Table 3). These parameters of ILD progression were all significantly more frequent in the CTD-ILD than in the IPAF group at the post-hoc pair-wise analysis.

### 3.3. Predictive Factors for ILD Progression at T1

As shown in Table 4, having a UIP pattern or a more than doubled RF were independently associated with ILD progression on multivariate analysis sex- and age-adjusted. On the other hand, having an IPAF independently predicted a better prognosis at T1. 

Interestingly, at last follow-up visit, of the four IPAF patients with a UIP pattern, two (50%) remained stable on HRCT and PFTs, one died for an acute exacerbation, and one developed a progressive ILD secondary to a definite SS.

## 4. Discussion

In our study, we retrospectively collected clinical data of ILD patients initially referred by the pulmonologist to the rheumatologist for a suspected SAD, and then followed-up by both these specialists. Several studies suggested that the contribution of the rheumatologist in the MDT discussion is essential to establish a confident diagnosis of ILD, not only at the onset but also during the follow-up [36,37]. For example, in Levi et al., patients initially classified as idiopathic pulmonary fibrosis (IPF) or hypersensitivity pneumonia received a modified diagnosis in favor of a SAD in nearly one-third of cases after rheumatological assessment [38]. 

Our cohort of IPAF patients presented similar features to other case series. Among our IPAF patients, nearly 75% were smokers or former smokers, and there was a slight female predominance (54.1%). Raynaud phenomenon and arthritis/morning joint stiffness were confirmed as the most represented features in the clinical domain, whereas ANA positivity was the most frequent autoantibody positivity in the serological domain. Most studies reported similar data [29,37,39,40,41]. 

Regarding the morphological domain, the NSIP pattern was the most prevalent in our IPAF cohort (56.8%), and this was consistent with the NSIP frequency of 68.9% reported by Sambataro et al. [29]. To date, the prevalence of ILD patterns considerably varies among IPAF case series, with particular regard to the UIP pattern, present in a relatively high percentage in the cohorts of Oldham et al. (45%) and of Sebastiani et al. (44.2%) [40,41]. This variability might deeply impact the results of the studies about ILD progression, limiting the prognostic value of IPAF classification. 

Of our IPAF cohort, eight patients (21.6%) were initially classified as lung-dominant UCTD before the publication of IPAF criteria in 2015. All these patients presented enough criteria to be classified as IPAF after revision of the items, and similar data were reported by Kelly et al. [42]. As already suggested by Ferri et al., IPAF might represent the pulmonary variant of UCTD [43]. 

In this study, the rate of progression of IPAF patients into a specific CTD-ILD (10.8%) during a median follow-up of almost 3 years was also consistent with the Literature [29,39,44,45]. Interestingly, all but one of our four evolved patients showed autoantibodies highly specific for CTD (anti-CCP, anti-SSA, anti-Jo-1). This confirms that antibody specificity could influence IPAF evolution towards a specific CTD [44]. In particular, anti-tRNA synthetase antibodies were associated with a high rate of progression into a well-defined SAD [9,46,47]. 

IPAF criteria limit the inclusion of patients with a UIP pattern to those with both serological and clinical domains satisfied, since it was argued that the UIP pattern is not sufficiently associated with SAD [6]. However, it is the most common pattern in RA-ILD, long-standing SSc-ILD and vasculitis [48]. It is also frequently observed in ILD-onset SS and up to 20% of Idiopathic Inflammatory Myopathies (IIMs) [48]. According to some authors, IPF patients with an autoantibody of the IPAF criteria or ANCA positivity had a higher probability of CTD development and better survival outcomes compared to those without any positivity (the so-called “lone” IPF) [49,50]. In an effort to characterize patients with a UIP pattern and only one item of the IPAF criteria, Sambataro et al. proposed the term “usual” interstitial pneumonia with autoimmune features (UIPAF), and showed that they could progress towards SAD in 28.9% of cases [13]. In line with this evidence, two of our 11 UIPAF patients (18.2%) developed a SAD during the follow-up.

So far, another great issue regards IPAF prognosis. In our cohort, IPAF patients had a significantly lower probability of reaching the composite endpoint of ILD progression on PFTs or HRCT, and a lower need for supplementary oxygen. Having an IPAF was also independently associated with a lower probability of ILD progression on multivariate analysis. However, compared to IPAF, our CTD-ILD cohort had a roughly three times higher prevalence of UIP pattern, which was confirmed as an independent predictor of ILD progression [12,42,51,52]. To date, several studies have described a broad heterogeneity within the IPAF natural history, mainly for the different prevalence of the UIP pattern among the cohorts analyzed [42,53,54]. Therefore, a revision to the IPAF morphological domain is advisable to uniformly classify patients in terms of prognosis. A possible solution could be the total exclusion of UIP patients from the IPAF criteria and their inclusion into a “parallel” UIPAF classification. This distinction may also be useful for comparison studies with “lone IPF”. 

In our study, a more than doubled RF was an independent predictor of ILD progression. Its overall prevalence was 12.9% and was not statistically different among IPAF, CTD-ILD and UIPAF groups. Kang et al. reported a similar frequency of RF positivity (13.2%) among a cohort of 688 ILD patients, mainly encompassed by IPF [55]. RF was independently associated with all-cause mortality and ILD-related death in RA patients [56,57]. Furthermore, serum RF titer significantly correlated with the risk of RA-ILD [58]. 

In our IPAF cohort, as reported by other studies, no evidence of some items considered in the IPAF criteria was found (e.g., distal digital tip ulcerations, Gottron’s sign), probably for their rarity or the high specificity for CTD [29,40,59,60]. On the other hand, we detected some features not included in the IPAF domains. Of these, objective evidence of sicca syndrome was found in 3 IPAF patients (8.1%) at the moment of diagnosis and developed in two IPAF patients during the follow-up, changing the diagnosis in favor of SS. In line with our study, sicca syndrome was frequently recorded in a prospective cohort of 52 IPAF patients and developed in four patients during the follow-up [40]. 

Though not expressly included in the IPAF criteria, anti-Ro52 antibodies were present in around 20% of our IPAF patients, while ANCA (anti-MPO) were present in one patient. These autoantibodies were among the most commonly observed within an IPAF cohort by Jee et al. [61]. Both autoantibodies might be associated with ILD and poor outcomes [62,63,64,65].

Given these data, future revisions of the IPAF criteria may include sicca syndrome within the clinical domain, and anti-Ro52 and ANCA within the serological domain. Other potential candidates might be muscle weakness and myalgia. Though nonspecific, these symptoms were associated with a higher prevalence of MSA/MAA among ILD patients [11]. 

Finally, this is probably the first study to evaluate the prevalence of various comorbidities in IPAF patients. Compared to the CTD-ILD and UIPAF cohorts, IPAF patients had a significantly higher prevalence of arterial hypertension and left-sided heart failure, and a not-statistically significant of dyslipidemia and Obstructive Sleep Apnea Syndrome (OSAS). These cardiovascular comorbidities could have played a role in the pathogenesis of ILD, at least indirectly. Angiotensin Receptor Blockers (ARBs), commonly used to treat arterial hypertension and heart failure, were associated with subclinical ILD and increased risk of all-cause mortality among IPF patients [66,67]. Statins may cause ILD as a side effect, while moderate/severe OSAS could lead to subclinical ILD through alveolar epithelial injury and extracellular matrix remodeling [68,69]. Finally, recent evidence showed that IPF patients had an incidence of left-sided heart failure almost four times higher than COPD, a lung disease with similar risk factors [70]. Further studies are needed to confirm these data in IPAF patients. 

The present study has some limitations. Firstly, it was a retrospective observational study in a single center, involving a relatively small number of patients. Secondly, we did not stratify patients according to the type of treatment. However, treatment was not a prognostic factor in our cohort. In addition, ILD progression was not influenced by treatment for other disease manifestations since all our patients had a pulmonary onset.

## 5. Conclusions

In our study, in around one year, IPAF patients showed a lower prevalence of ILD progression than patients with an ILD-onset CTD or with a UIP pattern and only one feature of the IPAF domains (the so-called “UIPAF”). However, ILD progression was predicted by the UIP pattern, independently from the final diagnosis. Furthermore, a high titer of RF could help to identify those patients at risk of progression.

Our data confirmed that both IPAF and UIPAF patients could develop into a CTD-ILD and could present clinical and serological features not included in the ERS/ATS IPAF classification criteria, such as sicca syndrome. For this reason, IPAF criteria should be revised to improve their ability to recruit all ILD patients with features of autoimmunity. On the other hand, the collocation of UIP pattern into the classification needs to be revaluated, given its highly predictive independent value of ILD progression among different cohorts of IIPs. A separation of the UIP pattern into a “parallel” UIPAF classification might increase the prognostic value of both IPAF and UIPAF definitions. Future prospective studies about UIPAF patients are advisable since preliminary evidence suggests a probability of evolving towards a CTD-ILD and, possibly, a different prognosis compared to IPF.

## Figures and Tables

**Figure 1 medicina-59-00794-f001:**
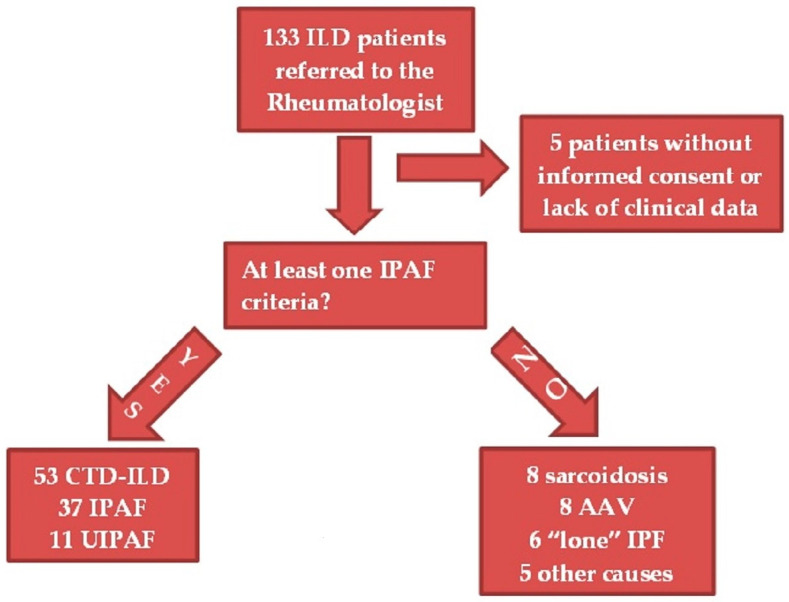
Patient classification. As “lone” IPF, we intended patients with IPF with clinical features due to a different condition (e.g., osteoarthritis) after rheumatological assessment (the so-called “lone” IPF). Legend: ILD, interstitial lung disease; IPAF, interstitial pneumonia with autoimmune features; CTD-ILD, connective tissue disease—interstitial lung disease; IPF, idiopathic pulmonary fibrosis; UIPAF, “usual” interstitial pneumonia with autoimmune features; AAV, anti-neutrophil cytoplasmic antibody (ANCA)-associated vasculitis.

**Figure 2 medicina-59-00794-f002:**
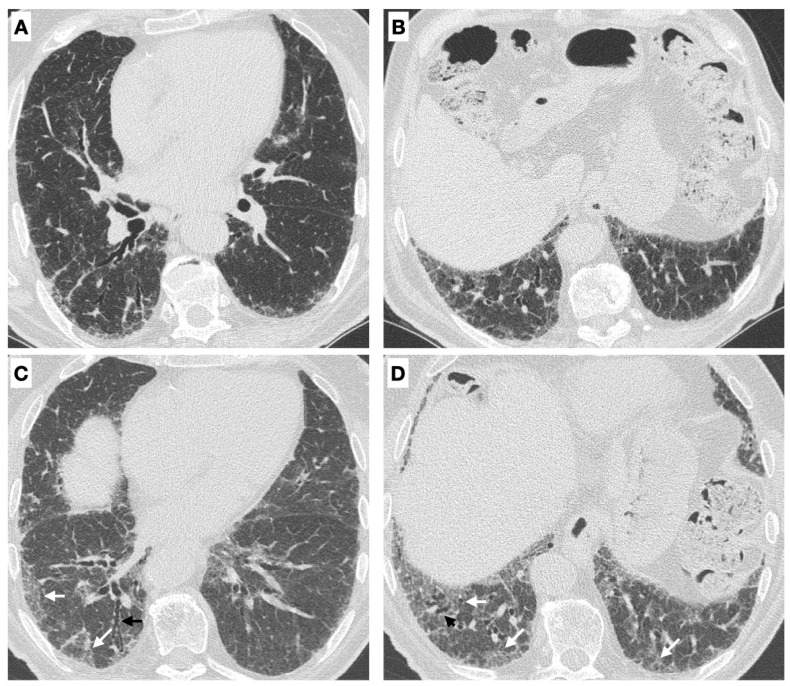
(**A**,**B**) Axial high resolution CT images show a subpleural irregular interlobular septal thickening and subtle ground glass opacities in the lower lobes. (**C**,**D**) A chest CT scan performed 1 year later demonstrated the radiological disease progression due to the increase in extent and severity of irregular inter and intralobular septal thickening (white arrows) and traction bronchiectasis/bronchiolectasis (black arrows).

**Figure 3 medicina-59-00794-f003:**
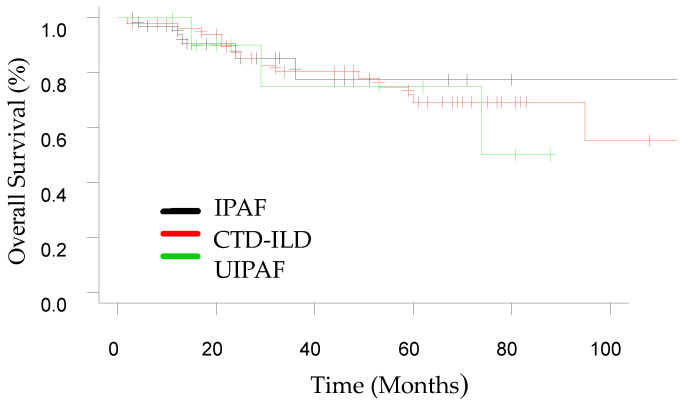
Comparison of survival curves of patients with IPAF, CTD-ILD and UIPAF.

**Table 1 medicina-59-00794-t001:** Clinical characteristics of IPAF, CTD-ILD, and UIPAF patients at diagnosis.

					Post-Hoc Analysis		
	IPAF	CTD-ILD	UIPAF	*p*-Value	IPAF vs. CTD-ILD	IPAF vs. UIPAF	CTD-ILD vs. UIPAF
No. of subjects	37 (36.6)	53 (52.5)	11 (10.9)				
Age (years)	71 (62.5–75.5)	69 (60.5–74)	68 (66–79)	0.64			
Female	20 (54.1)	37 (69.8)	8 (72.7)	0.27			
Never-smokers	13 (35.1)	23 (43.4)	4 (36.4)	0.77			
Months from symptoms’ onset	15 (7–39.5)	11 (6–34.5)	12 (6–17)	0.64			
Systemic hypertension	31 (83.8)	30 (56.6)	5 (45.5)	**<0.01**	**0.011**	**0.017**	0.526
Left-sided heart failure	9 (24.3)	3 (5.7)	2 (18.2)	**0.03**	0.024	1.00	0.201
Type 2 diabetes	7 (18.9)	10 (18.9)	2 (18.2)	1			
Hypercholesterolemia	17 (45.9)	17 (32.1)	4 (36.4)	0.41			
BMI (kg/m^2^)	27.5 (24.8–31)	25.6 (22.9–29)	25.5 (24–28.9)	0.09			
OSAS	6 (16.2)	3 (5.7)	1 (9.1)	0.25			
GERD	12 (32.4)	17 (32.1)	5 (45.5)	0.68			
Concurrent autoimmune disease	12 (32.4)	19 (35.8)	0 (0)	**0.04**	0.824	0.041	**<0.01**
Autoimmune thyroiditis	7 (18.9)	11 (20.8)	0 (0)				
Pulmonary hypertension (PH)	12 (32.4)	13 (24.5)	2 (18.2)	0.61			
Pre-capillary PH	3 (8.1)	6 (11.3)	0 (0)				
ARDS at onset	2 (5.4)	3 (5.6)	0 (0)	1			
Corticosteroid therapy	31 (83.8)	49 (92.5)	8 (72.7)	0.12			
With an immunosuppressant	18 (48.6)	26 (49.1)	1 (9.1)	**0.04**	0.970	0.032	0.018
With an anti-fibrotic agent	4 (10.8)	6 (11.3)	8 (72.7)	**<0.01**	0.940	**<0.01**	**<0.01**

Data are presented as number (%) or median (I-III quartiles) unless otherwise indicated. Bold font indicates statistical significance. Post-hoc tests significance threshold: *p* < 0.017. Legend: BMI, body mass index; OSAS, obstructive sleep apnea syndrome; GERD, gastroesophageal reflux disease; ARDS, acute respiratory distress syndrome.

**Table 2 medicina-59-00794-t002:** ILD pattern and PFTs of IPAF, CTD-ILD and UIPAF patients at diagnosis.

					Post-Hoc Analysis		
	IPAF	CTD-ILD	UIPAF	*p*-Value	IPAF vs. CTD-ILD	IPAF vs. UIPAF	CTD-ILD vs. UIPAF
Pattern							
UIP	4 (10.8)	17 (32.1)	11 (100)	**<0.01**	0.023	**<0.01**	**<0.01**
NSIP	21 (56.8)	23 (43.4)	0 (0)	**<0.01**	0.212	**<0.01**	**<0.01**
OP	4 (10.8)	2 (3.8)	0 (0)	0.3			
ILD + Multi-compartment involvement ꝉ	7 (18.9)	10 (18.9)	0 (0)	0.35			
Other patterns	1 (2.7)	1 (1.9)	0 (0)	1			
PFTs							
FEV1 (% predicted)	83 (76–102.5)	84 (75–101)	92 (76–115)	0.55			
FVC (% predicted)	86 (73–97)	91 (80–110)	94 (69–106)	0.3			
DLCO (% predicted)	61 (49–70)	49.5 (34.8–75)	46 (36.5–60.5)	0.36			
DLCO <35% and/or FVC <50%	5 (13.5)	14 (26.4)	3 (27.7)	0.28			
6MWT							
Distance (m)	495 (378–540)	459 (394–539)	461 (351–500)	0.53			
Lowest SpO2 (%)	92 (89–96)	91 (83–96)	86 (76–93)	0.12			

Data are presented as median (I-III quartiles) or number (%), unless otherwise indicated. Bold font indicates statistical significance. Post-hoc tests significance threshold: *p* < 0.017. ꝉ ILD and unexplained airways, vascular, pleural, or pericardial abnormalities, defined by the ERS/ATS IPAF classification criteria (see Text for further information). Legend: UIP, usual interstitial pneumonia; NSIP, nonspecific interstitial pneumonia; OP, organizing pneumonia; ILD, interstitial lung disease; PFTs, pulmonary function tests; FEV1, forced expiratory volume in one second; FVC, forced vital capacity; DLCO, diffusing capacity of the lungs for carbon monoxide; 6MWT, 6-min walk test; SpO2, peripheral capillary oxygen saturation.

**Table 3 medicina-59-00794-t003:** ILD progression at 12 ± 3 months from the diagnosis (T1).

					Post-Hoc Analysis		
	IPAF	CTD-ILD	UIPAF	*p*-Value	IPAF vs. CTD-ILD	IPAF vs. UIPAF	CTD-ILD vs. UIPAF
No. of subjects	35 (36.1)	51 (52.6)	11 (11.3)				
PFTs							
progression	9 (26.5)	22 (43.1)	7 (63.6)	0.07			
improvement	9 (25.7)	9 (17.6)	0 (0)	0.14			
stable	16 (47.1)	20 (39.2)	4 (36.4)	0.72			
DLCO < 35% and/or FVC < 50%	4 (11.4)	20 (39.2)	3 (27.3)	**0.01**	**<0.01**	0.337	0.516
HRCT							
progression	6 (17.1)	18 (35.3)	5 (45.5)	0.09			
improvement	9 (24.3)	4 (7.8)	0 (0)	**0.02**	0.029	0.087	1.00
stable	19 (54.3)	29 (56.9)	6 (54.5)	0.95			
Supplementary oxygen	3 (8.6)	16 (31.4)	4 (36.4)	**0.01**	**0.017**	0.050	0.735
At least one exacerbation	7 (20.0)	17 (33.3)	2 (18.2)	0.35			
Composite endpoint of ILD progression	11 (32.3)	30 (58.8)	8 (72.7)	**0.02**	**0.017**	0.033	0.461

Data are presented as number (%). On PFTs, ILD progression was defined by an absolute decline in forced vital capacity (FVC) of ≥5% predicted or in DLCO of ≥10% predicted. On HRCT, ILD trend (progressed, improved or stable) was established through a semi-quantitative analysis by a blinded thoracic radiologist. Patients with a PFTs and/or HRCT progression were included in the composite endpoint of ILD progression. Deceased patients were excluded. Bold font indicates statistical significance. Post-hoc tests significance threshold: *p* < 0.017. For more information see “Materials and methods”. Legend: PFTs, pulmonary function tests; DLCO, diffusing capacity of the lungs for carbon monoxide; FVC, forced vital capacity; HRCT, high resolution computed tomography.

**Table 4 medicina-59-00794-t004:** Factors predicting the composite endpoint of ILD progression at 12 ± 3 months from the diagnosis (T1) in the total ILD cohort, as determined by logistic regression analysis.

	Univariate		Multivariate	
Characteristics	OR (95% CI)	*p*-Value	OR (95% CI)	*p*-Value
Age (years)	1.00 (0.97–1.04)	**0.91**	0.98 (0.94–1.03)	0.53
Female sex	0.81 (0.34–1.88)	**0.62**	0.96 (0.34–2.74)	0.94
Never-smoker	0.85 (0.37–1.95)	0.71		
Pulmonary hypertension	2.36 (0.92–6.48)	0.12		
ARDS at onset	1.14 (0.13–9.83)	0.91		
IPAF	0.30 (0.12–0.72)	**<0.01**	0.28 (0.09–0.79)	**0.02**
UIP pattern	4.31 (1.73–11.7)	**<0.01**	3.80 (1.33–11.9)	**0.01**
ANA	0.76 (0.33–1.22)	0.51		
RF ≥ 2 times above the upper limit	5.77 (1.41–39.0)	**0.03**	9.13 (1.8–53.3)	**0.02**
Anti-CCP	0.96 (0.17–5.41)	0.96		
Anti-Ro (SSA)	0.48 (0.14–1.51)	0.22		
Anti-Scl-70	0.96 (0.04–24.7)	0.98		
Anti-tRNA synthetase	0.68 (0.21–2.13)	0.51		
Anti-PM/Scl	0.70 (0.13–3.36)	0.65		
Mechanic’s hand	0.47 (0.21–5.06)	0.54		
Arthritis or joint stiffness ≥ 60 min	1.02 (0.45–2.30)	0.97		
Palmar teleangiectasia	2.06 (0.76–6.03)	0.17		
Raynaud phenomenon	1.74 (0.72–4.35)	0.23		
Sicca syndrome	0.49 (0.19–1.23)	0.14		
Corticosteroid therapy	1.81 (0.56–6.41)	0.33		
With immunosuppressant	1.00 (0.45–2.26)	0.98		
With an anti-fibrotic agent ꝉ	9.00 (2.32–59.8)	**<0.01**		

Data are presented as Odds Ratio (OR) and 95% Confidence Interval (CI). The composite endpoint of ILD progression is characterized by an absolute decline in FVC ≥5% and/or in DLCO ≥10% predicted and/or deterioration on HRCT. Multivariate analysis was adjusted for sex and age and included data with a *p* < 0.1 on univariate analysis (bold font). ꝉ Therapy with an-fibrotic agent was excluded from multivariate analysis, given the high association with UIP pattern (phi = 0.59; *p* < 0.01). Legend: ARDS, acute respiratory distress syndrome; IPAF, interstitial pneumonia with autoimmune features; UIP, usual interstitial pneumonia; ANA, antinuclear antibody; anti-CCP, anti-Cyclic Citrullinated Peptide antibody.

## Data Availability

Clinical data about patients are collected in a database with less than 1% missing data. Only the authors have full access to them because of privacy restrictions. None of the raw data has been made available in any public repository.

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
