# Peer review of "Clinical Characterization and Predictive Factors for Progression in a Cohort of Patients with Interstitial Lung Disease and Features of Autoimmunity: The Need for a Revision of IPAF Classification Criteria"

_medicina, 2023, doi:10.3390/medicina59040794_

Round 1
Reviewer 1 Report
This is a retrospective paper regarding disease progression in a cohort of patients with non-IPF ILD with features of autoimmune disease. The paper is well-written, and the conclusions are clear and well-illustrated. However, revision of some sentences is mandatory. In addition, the criteria for disease progression used in this work are not the recently published ATS/ERS criteria.
In the abstract: “IPAF patients had a lower prevalence of UIP pattern compared to CTD- 32 ILD and, as expected, UIPAF (10.8% vs 32.1% vs 100%, p<0.01).” Please remove “as expected, UIPAF”. It is clear that UIPAF patients have a prevalence of 100% of UIP pattern.
“Arthritis was considered a clinical feature of IPAF only if unexplained by other conditions, while polyarticular morning stiffness ≥1 hour if associated with elevation of Erythro-Sedimentation Rate (ESR) or C Reactive Protein (CRP)” Please rewrite this sentence because the meaning is unclear
“To all patients, we performed spirometry, diffusing capacity of the lungs for carbon monoxide (DLCO) test, and Six Minutes Walking Test (6MWT) according to specific guidelines”.
Please modify the sentence in: “ we performed spirometry, diffusing capacity of the lungs for carbon monoxide (DLCO) test, and Six Minutes Walking Test (6MWT) in all patients, according to specific guidelines”
“Since Antisynthetase Syndrome (AS) is not currently classifiable by validated classification criteria, we decided to consider for IPAF those patients with anti-tRNA synthetase antibodies positivity plus a non-UIP pattern, for a definite diagnosis those with at least one of the other manifestations of disease, such as myositis or arthritis”.
Please rewrite as follows: Since Antisynthetase Syndrome (AS) is not currently classifiable by validated classification criteria, we decided to consider as IPAF those patients with anti-tRNA synthetase antibodies positivity plus a non-UIP pattern. The patients with at least one of the other manifestations of the disease, such as myositis or arthritis, were classified as having a definite diagnosis of AS.
1. Deterioration of lung function on PFTs, defined by a decline in forced vital capacity (FVC) of ≥10% predicted or in DLCO of ≥15% predicted, in accordance with other studies".
Recent guidelines addressing the definition of progression for progressive pulmonary fibrosis have been published. Currently, new criteria for physiological progression have been established. I suggest reclassifying physiological progression according to the following document: Idiopathic Pulmonary Fibrosis (an Update) and Progressive Pulmonary Fibrosis in Adults. An Official ATS/ERS/JRS/ALAT Clinical Practice Guideline
Why symptom progression was not assessed? I suggest including, if possible, this criterion.
Reviewer 2 Report
The paper provides a retrospective analysis of clinical data of interstitial lung disease (ILD) patients with a focus on patients who satisfy the criteria for IPAF and UIPAF. The criteria of UIPAF is somewhat arbitrary and the need to group a subset of patients into this category is highly debatable, if this is truly just chance that the presence of an autoimmunity factor is truly relevant for their condition or just chance. Here we see that UIP pattern obviously is the main predictor for progression of disease, which further supports the arbitrarily of such a classification.
1. The paper, particularly the introduction, must be extensively revised for language and clarity of the definitions. It is not organized in a linear way and difficult to understand if no background on the topics.
2. The authors could use more concise language to avoid repetition and simplify the text.
3. The authors could consider breaking up some of the longer paragraphs into shorter, more focused paragraphs to improve readability and comprehension.
4. The authors must consider adjusting their level of significance for performance of multiple tests (e.g. Bonferroni or similar). They perform 10-20 comparisons in a single table and report comparisons as significant, although if proper adjustments were made almost none would be considered relevant.
